# Effect of 3*d* Transition Metal Atom Intercalation Concentration on the Electronic and Magnetic Properties of Graphene/MoS_2_ Heterostructure: A First-Principles Study

**DOI:** 10.3390/molecules28020509

**Published:** 2023-01-04

**Authors:** Feng Wu, Zijin Wang, Jiaqi He, Zhenzhe Li, Lijuan Meng, Xiuyun Zhang

**Affiliations:** 1Department of Physics, Yancheng Institute of Technology, Yancheng 224051, China; 2College of Physics Science and Technology & Microelectronics Industry Research Institute, Yangzhou University, Yangzhou 225002, China

**Keywords:** graphene/MoS_2_ heterostructure, transition metal, intercalation concentration, electronic structure

## Abstract

The electronic and magnetic properties of graphene/MoS_2_ heterostructures intercalated with 3*d* transition metal (TM) atoms at different concentrations have been systematically investigated by first principles calculations. The results showed that all the studied systems are thermodynamically stable with large binding energies of about 3.72 eV–6.86 eV. Interestingly, all the TM-intercalated graphene/MoS_2_ heterostructures are ferromagnetic and their total magnetic moments increase with TM concentration. Furthermore, TM concentration-dependent spin polarization is obtained for the graphene layer and MoS_2_ layer due to the charge transfer between TM atoms and the layers. A significant band gap is opened for graphene in these TM-intercalated graphene/MoS_2_ heterostructures (around 0.094 eV–0.37 eV). With the TM concentration increasing, the band gap of graphene is reduced due to the enhanced spin polarization of graphene. Our study suggests a research direction for the manipulation of the properties of 2D materials through control of the intercalation concentration of TM atoms.

## 1. Introduction

In the past decades, atomic layer-thick two-dimensional (2D) materials have been attracting tremendous attention due to their extraordinary properties and possible application in diverse fields. On the other hand, the inherent shortcomings of 2D materials in certain areas greatly limit their application in certain fields. As a star material, graphene (G) has been found to have high electron mobility, thermal conductivity, light transmission, and much more [1]; however, the lack of bandgap and magnetism largely limits graphene’s application in nanoelectronics. In addition to graphene, 2D transition metal dichloride (TMD) also appears to have attractive properties for various applications, such as use in the fabrication of optoelectronics, transistors, etc. [2]. For example, MoS_2_ is a direct bandgap semiconductor with a very high switching rate, but its electron mobility is as low as 200 cm^−2^ V^−1^ s^−1^, which greatly hinders its use in the development of electronic devices [3]. Phosphorene is found to have a moderate direct band gap and relatively high hole mobility [4]; nevertheless, it is very susceptible to the environment and degrades over time, which greatly hinders its widespread use in electronic and optoelectronic devices [5].

As one solution, the construction of 2D van der Waals (vdW) heterostructures by stacking different monolayers provides an efficient way of achieving more intriguing properties for a wide range of applications. Taking graphene as an example, the band gap can be opened when it is adsorbed on other 2D materials, such as MoS_2_ [6], h-BN [7], and borophene [8]. In G/WS_2_ heterostructures, the weak spin–orbit coupling of graphene can be enhanced by the proximity effect of WS_2_ [9]. When stacked with *h*-BN, graphene displays significantly enhanced electron mobility, which allows the fractional quantum Hall effect of graphene to be observed [10]. Despite this, there is still much room for improvement in the properties of graphene. For example, the opened band gap of graphene in the above heterostructures is too small to meet the requirements of electronic device application. Moreover, the heterostructure will not have magnetism once the heterogeneous monolayers are no longer magnetic.

To date, various approaches have been developed to further manipulate the properties of 2D vdW heterostructures, such as creating point defects [11,12], doping impurity atoms [13,14], applying electric fields [15], exerting external strains [16,17], and intercalating metal atoms [18,19]. Among them, the intercalation of metal atoms in the interlayer gap has become a promising way of designing the physical properties of 2D vdW heterostructures [20,21,22,23], which has the advantage of not disrupting the structure of the monolayer [24]. These intercalators can act as bridges between the monolayers separated by vdW gaps, effectively enhancing interlayer interactions and influencing material properties through charge transfer, band gap engineering, phonon scattering, and so on [25,26,27,28]. For example, Ca intercalation in bilayer graphene has been shown to induce superconductivity, which is not present in intrinsic graphite [29]. Furthermore, Li intercalation has been shown to tune conductivity in several layers of MoS_2_ [30] and graphene/MoX_2_ (X = S, Se) [31] heterostructures by more than two to three orders of magnitude. Furthermore, a 3*d* transition metal (TM) atom-intercalated bilayer graphene [32,33], bilayer borophene [34], G/MoS_2_ [35], G/WS_2_ [35], G/WSe_2_ [36], and defective G/WSe_2_ [36] were found to introduce new electronic and magnetic properties. It is noted that the properties of intercalation compounds should be closely related to the concentration of the intercalated atoms, which is easy to control in experiments [20]. Nevertheless, the influence of intercalation concentration on 2D vdW material properties is still unclear.

Taking the nonmagnetic G/MoS_2_ heterostructure as an example, we intercalated magnetic 3*d* TM atoms at different concentrations into the graphene and MoS_2_ interlayer gap and attempted to tune the electronic and magnetic properties of it. Our results show that all the studied systems were thermodynamically stable with large binding energies of about 3.72 eV–6.86 eV. With increases in TM concentration, the binding energies were decreased to some extent. Due to the spin polarization of 3*d* TM atoms, all the studied systems were found to be ferromagnetic. Furthermore, the graphene layer and MoS_2_ layer were spin-polarized because of the charge transfer between the TM atoms and the layer. Moreover, a significant band gap was opened in graphene, which decreased with the TM concentration increase owing to the enhanced spin polarization of graphene.

## 2. Results and Discussion

First, we explored the structures and electronic properties of single TM atom-intercalated G/MoS_2_, 1TM@(G/MoS_2_) (TM = V, Cr, Mn, Fe). Three types of intercalated sites for locating TM atoms were tested for all the 1TM@(G/MoS_2_) systems (see Appendix A in the Appendix A): (i) a H-Mo site, in which the intercalated TM atom sits on the top site of a Mo atom and under the hollow site of graphene; (ii) a H-C site, in which the intercalated TM atom sits on the hollow site of MoS_2_ and under the C atom; and (iii) a H-H site, in which the intercalated TM atom sits between the hollow site of MoS_2_ and graphene. Our results proved that the single TM atom in the interlayer favors sitting in the H-Mo site, and the optimized structures of 1TM@(G/MoS_2_)s (TM = Ti, V, Cr, Mn) are shown in Figure 1a–d. Clearly, no significant distortion was found for all the studied systems. The distances of TM atoms to the graphene layer and MoS_2_ layer were 1.80/1.40 Å, 1.83/1.44 Å, 1.78/1.41 Å, and 1.96/1.33 Å for TM = Ti, V, Cr, and Mn, respectively.

To determine the structural stabilities of these 1TM@(G/MoS_2_)s (*n*TMs), we calculate the binding energy (*E*_b_) of the TM atom relative to the graphene layer and MoS_2_ layer using the following formula:*E*_b_ = [*E*_G_ + *E*_MoS2_ + *nE*_TM_ − *E_n_*_TM@(G/MoS2)_]/*n*(1)
where *E*_G_, *E*_MoS2_, *E*_TM_, and *E_n_*_TM@(G/MoS2)_ are the energies of the graphene layer, MoS_2_ layer, TM atom, and *n*TM@(G/MoS_2_), respectively, and *n* is the number of TM atoms. As shown in Figure 1e, the binding energy of these 1TM@(G/MoS_2_)s is quite large, ~4.61 eV–~6.89 eV, which is larger than that of individual TMs adsorbed on graphene [37] or MoS_2_ [38] monolayers. Thus, our results suggest that all the 1TM@(G/MoS_2_)s are thermodynamically stable. Furthermore, we find that *E*_b_s are sensitive to the choice of TM elements, i.e., *E*_b_ with V and Fe atoms is larger than with Cr and Mn atoms. The relatively low stabilities for 1Cr@(G/MoS_2_) and 1Mn@(G/MoS_2_) can be attributed to the half-occupied feature of their outermost electrons (Cr: 3*d*^5^4*s*^1^ and Mn: 3*d*^5^4*s*^2^). It is known that isolated metal atoms tend to aggregate into clusters due to high surface free energy [39]. To assess the feasibility of TM atom agglomeration, we compare the cohesive energy (*E*_coh_) of TM atoms in their metal crystal and *E*_b_s. The positive energy difference (Δ*E* = *E*_coh_ − *E*_b_) (see Figure 1e) means that the single TM atom is energetically more favorable than in the bulk form and is less likely to aggregate into clusters between the graphene and MoS_2_ layer.

To explicitly elucidate the bonding characteristics of the *n*TM@(G/MoS_2_)s systems, we calculated the charge density difference (CDD) as defined below:(2)$$\Delta \rho =\rho [nTM@(G/MoS_{2})]-\rho [nTM]-\rho [G]-\rho [{\mathrm{MoS}}_{2}]$$
where *ρ*[·] is the charge density of the whole system, *n*TM atom, graphene layer, and MoS_2_ layer, respectively. The CDD plots of 1Cr@(G/MoS_2_) and 1Mn@(G/MoS_2_) are plotted in Figure 1g, in which the Cr (Mn) atom loses electrons while the G and MoS_2_ monolayers gain electrons. Detailed charges transferred to graphene and MoS_2_ monolayers in all the studied 1TM@(G/MoS_2_) heterostructures are summarized in Table 1; from TM = V to Fe, the charge transferred to the monolayer on both sides decreases gradually.

Figure 2a–d plots the partial density of states (PDOS) and spin density of these 2D 1TM@(G/MoS_2_) heterostructures. Interestingly, the band gap of graphene is opened in these systems, with a significant band gap of about 0.37 eV, 0.28 eV, 0.30 eV, and 0.094 eV for TM = V, Cr, Mn, and Fe, respectively, which is much larger than those of G/MoS_2_ heterostructures without intercalation [40,41]. Moreover, contrary to the nonmagnetic nature of the initial host [11], the intercalation of TM atom introduces magnetic properties to 1TM@(G/MoS_2_)s. For example, the magnetic moments of 1TM@(G/MoS_2_)s are 3.93 µ_B_, 4.70 µ_B_, 3.69 µ_B_, and 2.00 µ_B_ for TM = V, Cr, Mn, and Fe, respectively. In addition, the charge transfer from the TM to the two-sided monolayers causes spin polarization in the graphene (MoS_2_) monolayer, whose local magnetic moment is around 0.02(0.48), 0.004(0.20), 0.06(0.26), and 0.06(0.24) for TM = V, Cr, Mn, and Fe, respectively. Particularly, 1Fe@(G/MoS_2_) is a Dirac half-metal, having a Dirac cone in the majority-spin channel while exhibiting a 94 meV gap in the minority-spin channel (see Figure 2d and Appendix A).

Furthermore, to explore the effect of different TM ratios on the electronic and magnetic properties of these TM-intercalated G/MoS_2_ systems, we investigated the G/MoS_2_ heterostructures of two- and three-TM atomic intercalations *n*TM@(G/MoS_2_) (TM = V, Cr, Mn, Fe, *n* = 2, 3). For 2TM@(G/MoS_2_)s and 3TM@(G/MoS_2_)s, nine and six isomers with different TM atom arrangements were tested, respectively (see Appendix A). For the former, the lowest energy structure of the system with TM = V, Cr, Mn is that of two TM atoms sitting in two adjacent hollow positions in graphene along the zigzag direction (see Figure 3a–c), whereas for 2Fe@(G/MoS_2_), the two Fe atoms prefer to sit a bit further away, i.e., they sit in two adjacent hollow positions in graphene along the armchair direction. For the latter, similar conformations were found for the most stable 3V@(G/MoS_2_) and 3Mn@(G/MoS_2_), where the three TM atoms are located on three adjacent hollow sites of the graphene forming a “<”-shaped pattern (see Figure 3e,g). As for 3Cr@(G/MoS_2_), the shape of the three Cr atoms was found to be almost linear, with two of the end Cr atoms located at the edge sites of the graphene, leaving the middle Cr atom located at the hollow site (see Figure 3f). For 3Fe@(G/MoS_2_), two of the three Fe atoms sit far from each other, forming an “L” shape (see Figure 3h). Similar to 1TM@(G/MoS_2_)s, no significant structure deformation is found for the graphene layer or MoS_2_ layer. Except for 2(3)Fe@(G/MoS_2_) and 3Cr@(G/MoS_2_), with TM atoms sitting at a different height along the *z* axis, all the TM atoms are found to stay in one plane. Moreover, the G/MoS_2_ interlayer distances (*d*) of these *n*TM@(G/MoS_2_)s are found to increase with the number of TM atoms (see Table 1), i.e., *d*_1TM@(G/MoS2)_ < *d*_2TM@(G/MoS2)_ < *d*_3TM@(G/MoS2)_.

As shown in Table 1 and Figure 4a, the binding energies per TM atom for these 2TM@(G/MoS_2_)s and 3TM@(G/MoS_2_)s are 6.03/5.70 eV, 4.08/3.79 eV, 4.02/3.72 eV, and 5.23/5.15 eV for TM = V, Cr, Mn, and Fe, respectively, which are smaller than those of the 1TM@(G/MoS_2_)s. We note that the *E*_b_s per TM atom for these *n*TM@(G/MoS_2_)s is related to the ratio of intercalated TM atoms, i.e., the higher the TM ratio, the smaller the *E*_b_s per TM atom. Our results show that single TM atom intercalation is the most energetically favorable in all systems. This stability related to the TM ratio is consistent with the transferred charge between the TM atoms and the two face layers, i.e., ΔQ_1_ + ΔQ_2_ decreases as *n* increases from 1 to 3 (see Table 1 and Figure 4b,c). Furthermore, as *n* increases, the charge redistribution around the TM atoms is no longer equivalent to their different positional sites (see inset in Figure 4b,c).

The PDOS and spin density plots of these 2TM@(G/MoS_2_)s and 3TM@(G/MoS_2_)s are shown in Figure 5. Similar to 1TM@(G/MoS_2_)s, the systems intercalated with two or three TM atoms are magnetic, with their magnetic moments mainly contributed to by 3*d* orbitals of TM atoms. Our results show that the magnetic moments of 2TM@(G/MoS_2_)s and 3TM@(G/MoS_2_)s are nearly double or triple those of 1TM@(G/MoS_2_)s. For the former, they are around 5.81 µ_B_, 10.51 µ_B_, 6.83 µ_B_, and 5.12 µ_B_ for TM = V, Cr, Mn, and Fe, respectively, while for the latter, they are 9.64 µ_B_, 14.73 µ_B_, 10.63 µ_B_, and 8.26 µ_B_ for TM = V, Cr, Mn, and Fe, respectively. Moreover, the spin polarization of graphene and MoS_2_ monolayers increases as the number of TM atoms increases. On one hand, the local moments of graphene and MoS_2_ monolayers in 2TM@(G/MoS_2_)s/3TM@(G/MoS_2_)s are increased to 0.03(0.03)/0.02(0.33), 0.02(1.00)/0.14(0.58), 0.006(1.16)/0.10(1.29), and 0.05(0.29)/0.06(0.19) for TM = V, Cr, Mn, and Fe, respectively. On the other hand, the larger spin polarization reduces the band gap in graphene, as shown in Figure 5; the band gaps of graphene in 2TM@(G/MoS_2_)s/3TM@(G/MoS_2_)s are 0.30/0.21 eV, 0.21/0.0 eV, 0.24/0.16 eV, and 0.26/0.27 eV for TM = V, Cr, Mn, and Fe, respectively. Therefore, we can conclude that the electronic and magnetic properties can be manipulated by varying the ratio of intercalated TM atoms, which is feasible in experiments through control of electrochemical voltage [20].

## 3. Computational Methods

All the spin-polarized density functional theory (DFT) calculations were performed with the Vienna ab initio simulation package (VASP) [42]. The ion–electron interactions were described using the projected-augmented wave (PAW) method [43]. DFT-D_2_, a semi-empirical method, was used to consider the van der Waals (vdW) interactions [44]. The exchange-correlation potentials were obtained using generalized gradient approximation (GGA) as parameterized by Perdew, Burke, and Ernzerhof (PBE) [45]. The energy cut-off for the plane-wave basis set was set to 400 eV. The energy convergence threshold was 10^−6^ eV per unit cell, and the ionic force on all relaxed atoms was less than 0.01 eV/Å. A vacuum layer of about 15 Å was used along the *z* direction to avoid interactions between neighboring cells. To consider the 3*d* electron’s strong correlation effect, a GGA + U method [46] was adopted with U = 3 eV in accordance with previous studies [47,48]. The graphene/MoS_2_ heterostructure (G/MoS_2_) was constructed as periodic slabs with the lattice parameters **a** = **b** = 12.35 Å, in which a 5 × 5 slab of graphene and 4 × 4 slab of the MoS_2_ monolayer were employed. One, two, and three TM atoms were intercalated into the interlayer gap of G/MoS_2_ heterostructures, which is denoted as *n*TM@ (G/MoS_2_) (*n* = 1, 2, 3). A Monkhorst–Pack grid of 5 × 5 × 1 was used for geometrical optimization of all the systems, and much denser *k*-point grids of 15 × 15 × 1 were used to explore electronic properties.

## 4. Conclusions

In conclusion, the electronic and magnetic properties of TM-intercalated G/MoS_2_ heterojunctions were systematically studied. Our results revealed that all the *n*TM@(G/MoS_2_)s (TM = Ti, V, Cr, Mn, *n* = 1–3) are very stable and have large binding energies. On one hand, all the studied systems are ferromagnetic, and their magnetic moments increase with *n*. In addition, spin polarization was observed in the graphene layer and MoS_2_ layer, and the degree of polarization increased with TM concentration. On the other hand, a significant band gap is opened for graphene upon the introduction of TM atoms, and the band gap is found to reduce with *n* due to the enhanced spin polarization of graphene. These results show that intercalation at different concentrations is a powerful approach for manipulating the electronic and magnetic properties of 2D vdW heterostructures and is thus expected to be widely applicable to other 2D layer materials and beyond.

## Figures and Tables

**Figure 1 molecules-28-00509-f001:**
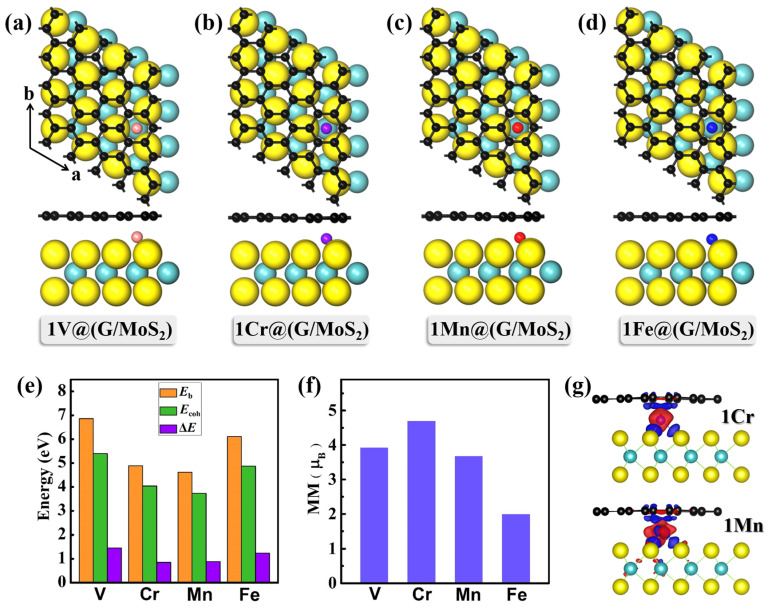
(**a**–**d**) The lowest energy structures of 1TM@(G/MoS_2_) (TM = Ti, V, Cr, Mn). Green, yellow, black, pink, purple, red, and blue balls represent Mo, S, C, V, Cr, Mn, and Fe atoms, respectively. (**e**) The binding energy (*E*_b_) of one TM atom between G and MoS_2_, the cohesive energy of TM atoms in their metal crystals (*E*_coh_), and the energy difference between them (Δ*E* = *E*_coh_ − *E*_b_). (**f**) Magnetic moments (MM) of 1TM@(G/MoS_2_). (**g**) The charge density difference (CDD) plots of 1Cr@(G/MoS_2_) and 1Mn@(G/MoS_2_).

**Figure 2 molecules-28-00509-f002:**
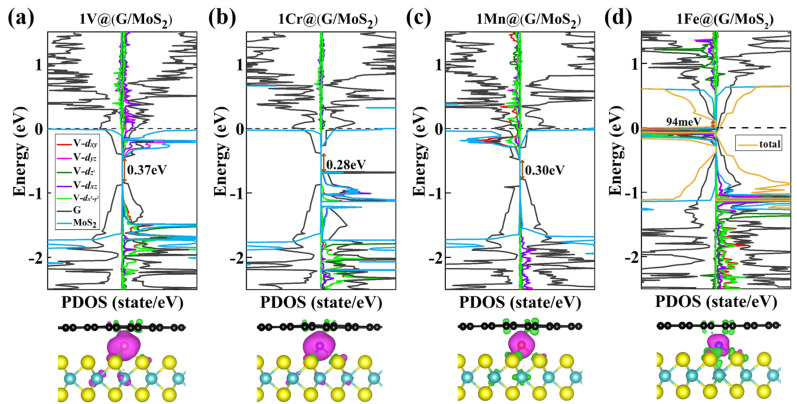
(**a**–**d**) Partial density of states (PDOS) and spin density plots for 1TM@(G/MoS_2_) (TM = Ti, V, Cr, Mn). The left and right panels of every PDOS plot are the minority- and majority-spin channels, respectively.

**Figure 3 molecules-28-00509-f003:**
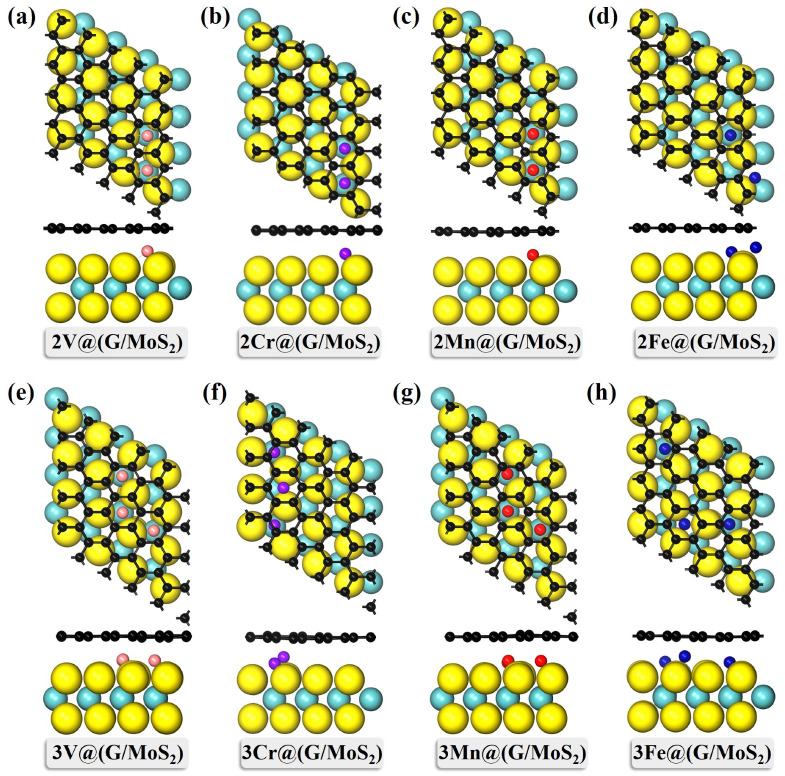
The lowest energy structures of (**a**–**d**) 2TM@(G/MoS_2_) and (**e**–**h**) 3TM@(G/MoS_2_) (TM = Ti, V, Cr, Mn).

**Figure 4 molecules-28-00509-f004:**
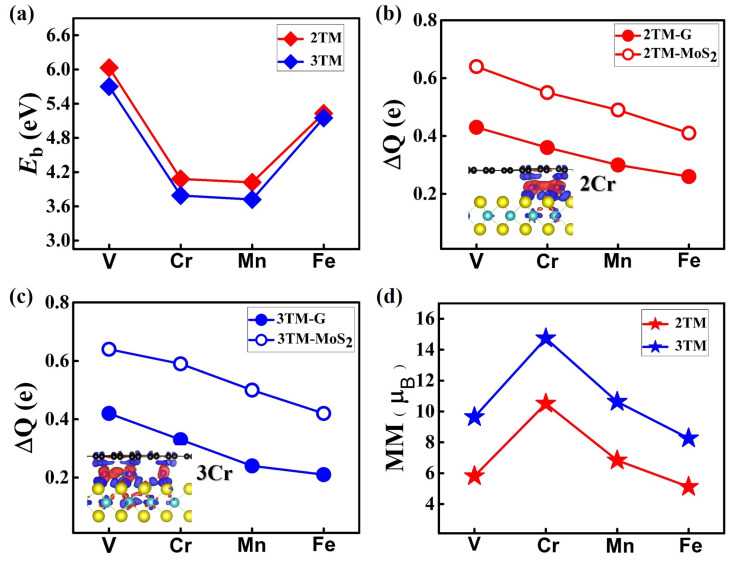
(**a**) The binding energy (*E*_b_) of two TM atoms and three TM atoms in-between G and MoS_2_. (**b**,**c**) The charges transferred from TM atoms to G and MoS_2_ for 2TM@(G/MoS_2_) and 3TM@(G/MoS_2_). (**d**) Magnetic moments (MM) of 2TM@(G/MoS_2_) and 3TM@(G/MoS_2_).

**Figure 5 molecules-28-00509-f005:**
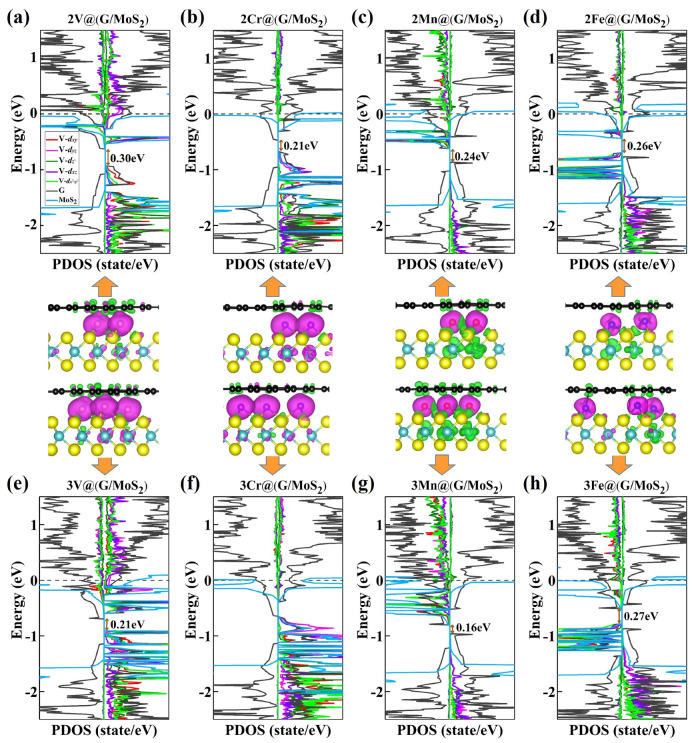
Partial density of states (PDOS) and spin density plots for (**a**–**d**) 2TM@(G/MoS_2_) and (**e**–**h**) 3TM@(G/MoS_2_) (TM = Ti, V, Cr, Mn).

**Table 1 molecules-28-00509-t001:** Distances between the graphene layer and MoS_2_ layer (*d*, Å), the binding energies (E_b_, eV), charges transferred from TM atoms to graphene (ΔQ_1_, e) and MoS_2_ (ΔQ_2_, e), and magnetic moments (MM, µ_B_).

	1TM@(G/MoS_2_)				2TM@(G/MoS_2_)				3TM@(G/MoS_2_)			
	*d*	*E* _b_	ΔQ_1_/ΔQ_2_	MM	*d*	*E* _b_	ΔQ_1_/ΔQ_2_	MM	*d*	E_b_	ΔQ_1_/ΔQ_2_	MM
V	3.20	6.86	0.46/0.66	3.93	3.53	6.03	0.43/0.64	5.81	3.59	5.70	0.42/0.64	9.64
Cr	3.27	4.89	0.41/0.65	4.70	3.74	4.08	0.36/0.55	10.51	3.80	3.79	0.33/0.58	14.73
Mn	3.19	4.62	0.38/0.57	3.69	3.52	4.02	0.30/0.49	6.83	3.64	3.72	0.24/0.50	10.63
Fe	3.29	6.11	0.12/0.56	2.00	3.52	5.23	0.26/0.41	5.12	3.57	5.15	0.21/0.42	8.26

## Data Availability

All relevant data are contained in the present manuscript.

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
