# Peer review of "Effect of 3d Transition Metal Atom Intercalation Concentration on the Electronic and Magnetic Properties of Graphene/MoS2 Heterostructure: A First-Principles Study"

_molecules, 2023, doi:10.3390/molecules28020509_

Round 1

Reviewer 1 Report

This is a good paper with basic results of theoretical calculations on the intercalation of transition metals into graphene/MoS2 heterostructures. I recommend the publication in Molecule, but it would be better if the authors can improve the manuscript based on the following points.

1. Abstract

It starts by "The structural, electronic and magnetic properties of 3d transition metal atom-intercalated 10 graphene/MoS2 heteostructures, nTM@(G/MoS2) (TM= V, Cr, Mn, Fe; n=1, 2, 3) "

The reader may not be able to understand what "nTM@(G/MoS2)" means here because "n" is not defined well in this sentence.  Is it the number of transition metal atoms in a 5x5 slab?  If so, it is not appropriate to use n in the abstract because it is hard to understand without knowing the computational detail. 

2. Introduction

I want more clarification on the motivation for choosing the graphene/MoS2 heterostructure as the parent material introduction part. Does it have a good lattice matching? Is the combination expected to have attractive properties based on their electronic state?

3. Spin-resolved PDOS

It is fascinating that the Dirac cone of graphene is spin polarization. This data recall the Dirac half metal described in [PHYSICAL REVIEW B 92, 201403(R) (2015)] expected in Mn-intercalated graphene/SiC. Is it possible that a similar Dirac half-metal-like state is realized in Fe-intercalated graphene/MoS2, except for the band gap?

Reviewer 2 Report

v  Authors have reported the interesting properties of graphene/MoS2  heterostructure. However, There are several approaches already reported on this material. Therefore, I suggest to emphasis the uniqueness and novelty of the present approach in detail.    

 v  What is the motivation of choosing V, Cr, Mn, Fe in this study

 v  Discussion section is unclear, such as effect,  accomplishment of this work and how this approach can be applied etc.

 v  Is it possible to apply this approach to other 2D materials such Ws2, Sns2, MoSe2 etc.

  v  I suggest to remove “structural”, since there were no structural properties in this work.

Reviewer 3 Report

The presented study considers timely problems of engineering interfacial physics in 2D materials, particularly their van der Waals heterostructures. In my opinion the manuscript is well-written and may be interesting for the community. However, there are few minor issues I would like to be addresses. Firstly, I can’t find clear and directly motivation for the conducted calculations. Please provide one and explain what is expected by performing the presented analysis? What Authors are looking for? Second, do Authors expect that intercalation may have an important impact on the potential barriers at the interface between 2D materials? If yes, what is the expected influence? Please describe briefly. The Authors may refer to ACS Appl. Mater. Interfaces 7 (2015)  25709 and Phys. Rev. B 97 (2018) 195315, for more details on potential barriers in transition metal dichalcogenides. Similarly, how the presented intercalation may influence charge transfer in such structures. See J. Appl. Phys. 117 (2015) 225101 for exemplary transport considerations in similar silicene/graphene heterostructures. Lastly how the presented intercalated structures compares to other sibling structures? I’m missing some context when the results are discussed in the conclusions. Authors are welcomed to improve their introduction in this regard as well.
